# The Antiviral, Anti-Inflammatory Effects of Natural Medicinal Herbs and Mushrooms and SARS-CoV-2 Infection

**DOI:** 10.3390/nu12092573

**Published:** 2020-08-25

**Authors:** Fanila Shahzad, Diana Anderson, Mojgan Najafzadeh

**Affiliations:** School of Life Sciences, University of Bradford, Bradford BD7 1DP, UK; fanilashahzad@hotmail.co.uk (F.S.); d.anderson1@bradford.ac.uk (D.A.)

**Keywords:** coronavirus, COVID-19, SARS-CoV-2, herbs, mushrooms, antiviral

## Abstract

The 2019 novel coronavirus, SARS-CoV-2, producing the disease COVID-19 is a pathogenic virus that targets mostly the human respiratory system and also other organs. SARS-CoV-2 is a new strain that has not been previously identified in humans, however there have been previous outbreaks of different versions of the beta coronavirus including severe acute respiratory syndrome (SARS-CoV1) from 2002 to 2003 and the most recent Middle East respiratory syndrome (MERS-CoV) which was first identified in 2012. All of the above have been recognised as major pathogens that are a great threat to public health and global economies. Currently, no specific treatment for SARS-CoV-2 infection has been identified; however, certain drugs have shown apparent efficacy in viral inhibition of the disease. Natural substances such as herbs and mushrooms have previously demonstrated both great antiviral and anti-inflammatory activity. Thus, the possibilities of natural substances as effective treatments against COVID-19 may seem promising. One of the potential candidates against the SARS-CoV-2 virus may be *Inonotus obliquus* (IO), also known as chaga mushroom. IO commonly grows in Asia, Europe and North America and is widely used as a raw material in various medical conditions. In this review, we have evaluated the most effective herbs and mushrooms, in terms of the antiviral and anti-inflammatory effects which have been assessed in laboratory conditions.

## 1. COVID-19 SARS-CoV-2 Infection

The novel coronavirus, SARS-CoV-2, causes severe acute respiratory syndrome and has quickly become a serious threat to public health [1]. Since the initial cases were reported in Wuhan, China, the virus has quickly spread globally, affecting more than 200 countries. As to date, over 20,046,642 cases have been identified with more than 734,525 deaths being reported [2]. Coronaviruses are enveloped positive-sense single-stranded RNA viruses, that primarily target the human respiratory system. The viruses have been shown to cause disease in both animals and humans [3]. Coronaviruses have either a round or elliptical form with an approximate diameter of 60–140 nm [4]. Coronaviruses belong to the Coronaviridae family, of which there are four subgroups: alpha (α), beta (ß), gamma (γ) and delta (δ). Of these subgroups, ß-coronaviruses tend to cause the most severe disease and fatalities in human populations [5]. Over the past two decades, two highly pathogenic ß-coronaviruses have been identified in humans, including severe acute respiratory syndrome (SARS-CoV-1) from 2002 to 2003, and the most recent Middle East respiratory syndrome (MERS-CoV) which was first identified in 2012 [6]. Based on its genomic structure and phylogenetic relationships, the novel coronavirus SARS-CoV-2 has also been identified as a ß-coronavirus. The coronavirus genome is comprised of approximately 30,000 nucleotides and is enclosed in a lipid envelope. A typical coronavirus contains around six open reading frames (ORFs) within its genome. Two-thirds of viral RNA, mainly located in the ORF 1a/b, encode for 16 non-structure proteins. The rest of the virus genome encodes for structural and accessory proteins associated with the virus [7]. Four main structural proteins are encoded by ORFs 10 and 11, these include a spike (S) protein, an envelope (E) protein, a membrane (M) protein, as well as the nucleocapsid (N) protein [8]. The N protein is bound to the virus single-positive strand RNA and allows the virus to hijack host cells. The N protein also coats the viral RNA genome and has also been shown to play an important role in viral replication and transcription. The M protein is thought to act as a central organiser for coronavirus assembly and is also the most abundant protein on the viral surface. The E protein is a membrane protein composed of approximately 76 to 109 amino-acids, and the protein plays an important role in virus–host cell interaction and virus assembly [9]. Despite displaying similarities with the other human ß-coronaviruses, SARS-CoV-2 possesses many differences in its genomic and phenotypic structure which greatly influences the pathogenesis of SARS-CoV-2 [8,10]. 

Coronaviruses have been described as zoonotic infections, with alpha and beta coronaviruses found primarily in mammals such as bats whereas gamma and delta are more common in pigs and birds. Additionally, investigations into the previous coronavirus outbreaks found that SARS-CoV1 was transmitted from bats to humans and MERS-CoV from dromedary camels to humans [5]. Much like its predecessors, SARS-CoV-2 has also proven to be successful in making its transmission from an animal host to humans. Many efforts have been made in order to identify the reservoir host or the intermediate host for the novel coronavirus. Despite the fact the exact origin of SARS-CoV-2 is yet to be identified, genomic analysis of SARS-CoV-2 has shown 88% similarity between two severe acute respiratory syndromes (SARS)-like coronaviruses derived from bats, thus demonstrating that SARS-CoV-2 could have evolved from a coronavirus of bat origin [3,10,11]. However, there have also been other reports linking SARS-CoV-2 to snakes and even pangolins [3]. 

The primary mode of human to human transmission of SARS-CoV-2 has been identified to be through respiratory droplets [12]. Once the virus has gained entry into a human host, reports have shown that SARS-CoV-2 infects cells using receptor-mediated endocytosis via the membrane-bound aminopeptidase angiotensin-converting enzyme II receptor (ACE2). Studies have shown that the primary target for this novel virus seems to be the lung alveolar epithelial cells, which ultimately results in the manifestation of respiratory symptoms [11]. Additionally, a recent study has shown that SARS-CoV-2 exhibits a 10-fold higher affinity for the ACE2 receptor, thus explaining its high transmission rate compared to that of both SARS-CoV1 [13]. 

COVID-19 symptoms appear after an incubation period of around 2–14 days. The duration from the onset of symptoms to death has ranged from 6 to 41 days. However, this duration is dependent on both the age of the patient and the status of their immune system [11]. Patients over 70 years of age seem to be more susceptible to this virus, probably as a result of a weaker immune system [14]. SARS-CoV-2 infection has been associated with many symptoms and clinical signs. However, based on hospitalised data, patients mainly exhibited symptoms of acute respiratory distress syndrome, associated with pyrexia, cough, fever and fatigue [14]. Additionally, one study found that approximately 80% of patients reported being asymptomatic or with mild symptoms (*n* = 19). The remaining 20% of cases, however, were more critical and severe [15]. COVID-19 has shown similar clinical presentations to that of its predecessors; however, in contrast to SARS-CoV and MERS-CoV, COVID-19 seems to have a lower fatality rate [3]. Despite there being limited information regarding the innate immune response of patients with COVID-19, most studies have shown that patients often develop lymphopenia [16]. A recent study investigating 99 patient cases in Wuhan demonstrated an increase in neutrophils (38%), lymphopenia (35%), increased IL-6 (52%) as well as an increase in C- reactive protein in 84% of cases [16]. Moreover, this increase in neutrophils and a decrease in lymphocytes have proven to directly correlate with both disease severity and fatality [16]. In addition, patients who had severe SARS-CoV-2 infection also presented with high levels of pro-inflammatory cytokines such as IL-7, IL-10, IL-2, G-CSF, MCP-1, MIP-1A and TNFα. These findings correlate with the clinical profile of SARS-CoV-1 and MERS-CoV, thus suggesting that lymphopenia and the presence of a cytokine storm contributed to the pathogenesis of SARS-CoV-2 [17].

Due to the increasing number of human casualties, research has focused on understanding the nature of the disease in order to develop effective treatments. Currently, no specific treatment has been identified for COVID-19. The race to develop a preventive vaccine is still ongoing, with many still in the early stages of clinical trials. Therefore, there is a dire need to develop an effective antiviral with enhanced efficacy for the prophylaxis and treatment of coronavirus infection. Various medications have now been identified and developed as a means for controlling and suppressing inflammatory crises such as steroids, nonsteroidal anti-inflammatory drugs and immunosuppressants [18]. In practice, the goal is to develop a drug with the minimum effective dose, which has increased efficacy. However, developing such medications often means that they are associated with adverse side effects such as ulceration, gastric irritation, angioedema, hepatic failure, headache, haemolytic anaemia, hyperglycaemia and immunodeficiency-related problems, as well as others [18]. Therefore, the use of natural medicinal products that are generally considered safe as a form of alternative therapy to increase the pharmacological response with the lowest degree of unwanted side effects is now being considered [18]. There have been many reports regarding the use of natural substances isolated from medicinal plants as effective treatments against viral infections. Lin et al. (2014) summarised the antiviral capabilities of herbal medicines against several viral pathogens such as coronavirus, coxsackie virus, hepatitis B virus, hepatitis C virus, herpes simplex virus (HSV), human immunodeficiency virus (HIV), influenza virus and respiratory syncytial virus (RSV). In regard to coronaviruses, this review highlighted that naturally occurring triterpene glycosides, known as saikosaponins, isolated from medicinal plants such as Bupleurum spp., Heteromorpha spp. and Scrophularia exhibited great antiviral activity against human coronaviruses (Table 1). These naturally occurring compounds were able to effectively prevent early stages of coronavirus infection by affecting viral attachment and cell penetration. Moreover, many natural compounds such as myricetin, scutellarein and phenolic compounds from Isatis indigotica and Torreya nucifera have been identified as natural inhibitors against the number of coronavirus enzymes, including nsP13 helicase and 3CL protease [19]. The water extract from Houttuynia cordata has also been identified as another anti-coronavirus natural medicine, as it has been observed to both inhibit the viral 3CL protease and block the viral RNA-dependent RNA polymerase activity, thus displaying various antiviral mechanisms against SARS-CoV1 [20]. Based on the data presented, this mini-review aims to review the antiviral and the anti-inflammatory effects of natural herbs and mushrooms against viral infections (Table 2) as well as to provide an insight into the possibilities of using such natural as effective treatments against COVID-19.

## 2. The Potential Use of Mushrooms and Herbs against SARS-CoV-2 Infection

Currently, no specific treatment for COVID-19 has been recognised; however, certain drugs have shown apparent efficacy in viral inhibition of the disease [15]. Using an artificial intelligence (AI) programme, a group of researchers found potential therapeutic candidates that could inhibit clathrin-mediated endocytosis and thus inhibit viral infection. Such drugs could be used as potential therapeutics against COVID-19 [21]. However, in the cases of such commercial medications, there is always an increased risk of the potential development of drug resistance, especially in the case of specific viral enzyme inhibitors. Therefore, there is an urgent need to discover novel antivirals that are cost-effective and exhibit enhanced efficacy for the management and control of viral infections when vaccines and standard therapies are unsuccessful [19]. Natural substances such as mushrooms and herbs have previously shown both great antiviral and anti-inflammatory activity and thus serve as excellent sources of novel antiviral treatments, hence the possibilities of natural substances as effective novel treatments against COVID-19 may seem promising [22].

## 3. Antiviral Properties of Herbs

There are many populations that prefer the use of natural products in place of commercial pharmaceutically developed products for treating and preventing medical illnesses. Many cultures around the world have historically relied on medicinal plants for primary care, with some continuing use till this date [23]. Herbs are plants which typically possess fragrant and aromatic properties and are usually used in many cultures to flavour food dishes; however, it is widely known that herbs are also used as part of natural medicines primarily in Asian countries such as China [24]. Herbal plants and purified natural products provide a rich supply for the development of novel antiviral compounds [19].

Prunella is a genus of perennial herbaceous plants in the Labiatae family. Approximately 15 species have been identified worldwide, most of which are distributed in the temperate regions of Europe and Asia. From this genus, the most extensively studied is *Prunella vulgaris* (PV) (selfheal) due to its several thousand-year histories as an antipyretic and antidotal herb used in traditional Chinese medicine [24,25]. The Prunella genus is known to contain triterpenoids and their saponins, phenolic acids, sterols and associated glycosides, flavonoids, organic acids, volatile oil and saccharides. Various pharmacological studies investigating Prunella have demonstrated the enhanced antiviral, antibacterial, anti-inflammatory, immunoregulatory, anti-oxidative and anti-tumour properties it possesses [26]. PV is a herbaceous plant that is commonly known as self-heal and heal all [27]. PV has been reported to exhibit various biological properties including anti-inflammation, antimicrobial and even anticancer. PV has been widely studied via the use of in vitro studies; as a result, approximately 200 compounds have reported to be isolated from PV, many of which have been characterised as triterpenoids, sterols and flavonoids, followed by coumarins, phenylpropanoids, polysaccharides and volatile oils [25]. In order to investigate the antiviral mechanism of action of PV against Ebola virus (EBOV) infection, Zhang et al. developed a sensitive EBOV- glycoprotein (EBOV-GP) pseudo-typed HIV-1-based vector system. Based on this system, scientists showed that an aqueous extract from PV called CHPV exhibited potent inhibitory effects on EBOV-GP pseudo-typed virus in various cell lines including human umbilical vein endothelial cells and human macrophages. Moreover, the results concluded that in VeroE6 cells, CHPV was able to efficiently block eGFP-expressing Zaire Ebola virus (eGFP-ZEBOV) infection. The inhibition rate almost reached 99.5% when the concentration of CHPV was 20 μg/mL and when it was consistently present in the culture medium. This inhibitory effect of CHPV was observed in a dose exhibited in a dose-dependent manner. Additionally, at a concentration level of 12.5 μg/mL, CHPV showed more than 80% inhibition of EBOV-GP-V and eGFP-EBOV infections. In the same study, a “time-of-addition” study was performed in order to investigate the CHPV anti-EBOV mechanism of action. The results obtained concluded that the inhibitory effect of CHPV occurred via binding directly to EBOV-GP-Vs and possibly suppressing virus attachment through targeting the viral GP protein, thus blocking the early viral events [28]. Previous studies investigating PV found that aqueous extracts from this herb had the potential to decrease the replication of herpes simplex virus (HSV) by directly preventing the virus to bind to cells as well as having the capability to inhibit HIV-1 infection by preventing viral attachment to CD4+ T cell receptor [29,30]. It has been previously suggested that the antiviral abilities of CHPV may be due to an anionic polysaccharide. Polysaccharides have the ability to bind to and from complexes with viral proteins, which can ultimately block virus entry into cells and thus prevent early infection. However, this polyanionic property of CHPV is yet to be determined [28]. Interestingly, the results from this study also confirmed that CHPV has the ability to enhance anti-EBOV activity of the anti-EBOV antibody (2G4) against EBOV-GP. This antibody is an important component of the two-antibody cocktail, ZMAb and ZMapp. Previous investigations have shown that 2G4 has enhanced efficacy in blocking EBOV infection both in in vitro and in vivo animal models [31,32,33]. The results from this study showed that combined use of 2G4 and CHPV at low concentrations achieved the same anti-EBOV-GP efficacy as a high concentration of 2G4 alone. Overall, from this study, it can be concluded that that CHPV has great anti-EBOV activity and has the potential to be developed as a novel antiviral approach against EBOV infection [28]. This natural Chinese herb merits further investigations as an antiviral due to the promising results observed in its antiviral properties. Upon further research, these antiviral properties may also show efficacy against SARS-COV-2. 

The spike of Prunella vulgaris, also known as Prunellae Spica, is often used for the prevention and remedy of various diseases, in traditional Chinese medicine. Prunellae Spica contains various bioactive compounds within its chemical composition including triterpenes, flavonoids, phenolic compounds and carbohydrates. These compounds have been associated with a variety of protective effects such as anticancer, anti-inflammatory, neuroprotective, immunosuppressive activity and anti-HIV activity [34]. In regard to Prunellae Spica antiviral effects, one study was successful in isolating a novel partially sulphated polysaccharide with a molecular weight of around 32 kDa (PSP-2B), from aqueous extracts of Prunellae Spica. PSP-2B is mainly composed of the sugars arabinose, galactose and mannose, with small amounts of uronic acids and glucose. In the same study, the antiviral effects of Prunellae Spica against herpes simplex virus (HSV) were investigated. Researchers found that PSP-2B strongly demonstrates activity against HSV, with an IC50 of approximately 69 and 49 μg/mL for HSV-1 and HSV-2, respectively. At the same time, when the concentration of PSP-2B was increased in a gradual manner to 1600 μg/mL, no cytotoxicity was observed. The results for this study further demonstrate the great antiviral potential that Prunella has against viral infections such as herpes simplex virus and raises the questions whether similar effects could be demonstrated on viral infections such as COVID-19 [35]. 

Garlic (*Allium sativum*) is an aromatic herbaceous plant that has been extensively used worldwide, especially in the Far East, for centuries, in many food dishes due to its appetising property, bitter taste and the flavour it gives to dishes. Despite the use of garlic being such a common practice, this particular plant is of great medical importance, as garlic has exhibited antibacterial, antiviral, antifungal and even antitumor effects [36]. With more than 200 chemical substances, garlic has the ability to protect the human body from many illnesses. Despite researchers arguing in order for garlic to be effective, it should be consumed fresh; some research has proven that the effects of garlic are consistent when it is cooked, with oils even providing better protection against oxidative stress and infections [37]. *A. sativum* is rich in alliin, allicin, ajoenes, vinyldithiins and flavonoids such as quercetin, all of which are sulphur-containing phytoconstituents [36]. Very little work has been done to investigate the antiviral properties of *A. sativum* but the experiments that have been conducted have shown that *A. sativum* is effective against influenza B, herpes simplex viruses, cytomegalovirus, rhinovirus, HIV, HSV type 1 and 2 and viral pneumonia [38,39]. It has been speculated that in the case of HIV, the ajoene acts by inhibiting the integrin-dependent processes, thus inhibiting the fusion of cells with HIV-infected cells, and ajoene was also able to inhibit HIV replication, further showing its antiviral activity [40]. Additionally, a single clinical trial reported that garlic may prevent the occurrence of the common cold, however, the data provided are insufficient. Moreover, it has been scientifically proven that garlic is effectively used in cardiovascular diseases by regulating blood pressure, with reducing effects on glycaemia and high blood cholesterol [37]. Taken together, the data show the beneficial effects extracts of garlic have and thus make it useful in medicine. However, further research is still required to determine the exact mechanisms and its potential as an antiviral agent. 

In addition to garlic, ginger, also known as *Zingiber officinalis*, has shown great promise as a medicinal agent [38]. Ginger is a commonly used spice that is rich in terpenes, polysaccharides, lipids, organic acids, raw fibres and phenolic compounds. It has been reported that the health benefits of ginger are mainly due to the phenolic compounds such as gingerols and shogao it possesses [41]. Many investigations have demonstrated that ginger possesses a range of biological activities such as antioxidant, anti-inflammatory, antimicrobial, anticancer, neuroprotective, cardiovascular protective and antiviral properties [41]. In 2016, researchers tested the antiviral effects of ginger against feline calicivirus, a surrogate of human norovirus. The results from this study showed that in addition to garlic, ginger extracts significantly inhibited the calicivirus which was in a dose-dependent manner, thus confirming the specific viral potency ginger possesses [42]. In addition, in a separate study, it was found that fresh ginger had the potential to inhibit the attachment and internalisation of the human respiratory syncytial virus (RSV) to both human lung cells and liver cells, in a dose-dependent manner. It was also found that the treatment with ginger stimulated the mucosal cells to increase sections on Interferon-beta (INF-β) secretions, which has antiviral effects and thus possibly contributed to the inhibition of the viral infection [43]. In another study conducted in Japan, the researcher investigated the antiviral potential of ginger against influenza A in vitro. This study concluded that extracts of ginger stimulated the production of TNF-α, which has previously been shown to act as the first line of defence against virus infections [44].

Furthermore, a study conducted by Rasool et al. looked at the activity of aqueous extracts of ginger and garlic on anti-avian influenza virus H_9_N_2_, in chick embryos. Results obtained showed that the aqueous extract of ginger exhibited antiviral activity at 10, 15, 20 and 25%, whereas in the case of garlic, antiviral activity was observed at 15, 20 and 25%. Moreover, MTT assays revealed that both plants displayed cytotoxicity, which was in a dose-dependent manner; however, extracts of ginger exhibited lower cytotoxicity in comparison to ginger. From this study, it can be concluded that aqueous extract of ginger showed great antiviral activity against H_9_N_2_ and was less cytotoxic to cells with a cell survival rate of more than 50% [45]. Thus, it can be assumed that ginger may have the potential to become a promising natural antiviral agent; however, in order for this to happen, further investigation is still required. 

## 4. Antiviral Properties of Mushrooms

Mushrooms are described as macrofungi which have unique fruiting bodies, these can either be underground fruiting bodies (hypogeous) or they can have fruiting bodies above the ground (epigeous). It is becoming widely known that different mushrooms possess a variety of biological and pharmacologically active molecules. Previous research has shown that bioactive components and extracts derived from mushrooms exhibit strong anticancer activities [46]. In addition, extracts from mushrooms have also been shown to display antibacterial, antiviral, anti-inflammatory, antiatherogenic and hepatoprotective effects. Thus, mushrooms have a great potential for use as successful antiviral treatments with a reduced chance of adverse side effects [22]. 

A study conducted in 2018, determined the structural characterisation of lentinan from *Lentinus edodes* mycelia (shiitake) as well as looking at the antiviral activity against infectious hematopoietic necrosis virus (IHNV). *Lentinus edodes* mycelia extract is a powder that is extracted from shiitake mushrooms known as *Lentinus edodes.* The novel lentinan (LNT-1) was extracted and purified via anion exchange and the structural characterisation was carried out by a range of experiments including gas chromatography-mass spectrometry and 1D-nuclear magnetic resonance spectroscopy. The results obtained displayed that LNT-1 had an overall molecular weight of 3.79 × 10^5^ Da. Structural characterisation of LNT-1 showed that it was a β—(1 → 3)—glucan backbone with—(1 → 6)—glucosyl side-branching units terminated by mannosyl and galactosyl residues. Additionally, the study found prominent antiviral activity against INHV, and the main antiviral mechanisms of LNT-1 were direct inactivation as well as inhibition of viral replication. As well as this, administration of LNT-1 also caused significant downregulation in the expression of pro-inflammatory cytokines such as TNF-α, IL-2 and IL-11, whilst also upregulating the expressions of IFN-1 and IFN-γ, cytokines that are known to induce antiviral, anti-proliferative and immunomodulatory effects [47]. Overall, the results indicate the antiviral activity of LNT-1 and its regulation of the innate immune response. As previously said, the innate immune response is a critical factor for COVID-19 disease severity and disease outcome. COVID-19 patients exhibit high titres of inflammatory cytokines and so the effects of LNT-1 should be considered on SARS-COV-2 [48]. 

Another species of mushroom that has shown promising antiviral effects is *Grifola frondosa,* (hen-of-the-woods, ram’s head and sheep’s head) a species of *Basidiomycotina.* This is an example of an edible mushroom that has been used in herbal medicine. In comparison to the shiitake mushroom, *Grifola frondosa* has a higher nutritional value. The major biologically active component of the mushroom is the β-glucan in the *G. frondosa* polysaccharide (GFP). Moreover, GFP has shown great anticancer potential with it being approved as a therapeutic drug for the treatment of cancer in China [49]. A previous study purified a novel antiviral protein extract GFAHP from *Grifola frondosa* using ammonium sulphate precipitation and DEAE ion exchange chromatography. GFAHP has a reported molecular weight of 29.5 kDa, and the N-terminal sequence of GFAHP consists of 11 amino acid peptides. This peptide sequence did not match any known amino acid sequences, thus indicating that GFAHP is likely to be a novel antivirus protein. This protein extract displayed great ability to inhibit in vitro replication of HSV type 1 (HSV-1). In murine models, higher concentrations of GFAHP, in particular at the doses of 125 and 500 μg/mL, strongly reduced the severity of blepharitis, neovascularisation and stromal keratitis induced by HSV-1(Gu et al., 2007). Gu et al. (2007) found that topical administration of the GFAHP protein extract to the cornea of mice caused a significant decrease in virus production. This study demonstrated that GFAHP was able to both directly inactivate HSV-1 and inhibit HSV-1 infiltration into Vero cells [50]. Additionally, in the separate study, D-fraction was extracted from *Grifola frondosa* (GF-D) and was used in combination with human IFN α-2b (IFN) in order to investigate the inhibitory effect of hepatitis b virus (HBV). Following analysis of HBV DNA and viral antigens, the results obtained showed that GF-D or IFN alone were able to inhibit HBV DNA in cells with a 50% inhibitory concentration (IC50) of 0.59 mg/mL for GF-D and 1399 IU/mL for IFN. Upon further analysis, researchers found that combined use of GFD and IFN synergistically inhibited HBV replication. When combined with 0.45 mg/mL GF-D, the IC50 for IFN was reported to be 154 IU/mL, thus suggesting that in combination, there was a 9-fold increase in antiviral activity. The results indicate the possibility of using GF-D and IFN combination therapy as a potentially effective therapy against hepatitis b virus infections [51]. In another study, the effects of GF-D were analysed on 35 HIV-infected patients. Following administration with GF-D, CD4+ cell counts, viral load measure, symptoms of HIV, status of secondary disease and sense of wellbeing were monitors in order to test HIV status of each individual. From 35 patients, 57% presented with an increase in CD4+ cell count, whereas 22% reported a decrease in cell count. In regard to viral load, different results were observed: 9 patients displayed an increase in viral load, whereas a decrease was observed in the viral load of 10 patients. Despite this, 85% of patients reported an increase in sense of wellbeing with regard to symptoms and also secondary diseases that are linked to HIV, further suggesting the positive impact this extract can have against viral diseases [52]. 

Abu-Serie et al. evaluated the antioxidant and the anti-inflammatory effects of Malaysian *Ganoderma lucidum* aqueous extract (GLE) and Egyptian *Chlorella vulgaris* ethanolic extract (CVE). The main finding of this study shows that GLE-CVE exhibited higher antioxidant and anti-radical effects in comparison to individual extracts. GLE-CVE also attenuated lipopolysaccharide-induced inflammation and oxidative stress in white blood cells, which occurred through the process of downregulating inflammatory mediators such as TNF-α, cyclooxygenase-2, nuclear factor kappa-beta (κβ) as well as the expression of inducible nitric oxide synthase. In addition, the combined extracts also exhibited great ability in enhancing the cellular antioxidant indices. These changes caused by GLE-CVE also led to suppression of cellular increase in nitric oxide and lipid peroxidation. This study also concluded that the combined extract had an antioxidant effect that was significantly greater than that of a commercial anti-inflammatory drug, dexamethasone. As oxidative stress and inflammation are two factors that are consistently linked to the pathogenesis of COVID-19, there may be a great possibility of this combined extract as an alternative treatment [53].

A potential candidate against the SARS-COV-2 virus may be *Inonotus obliquus* (IO), also known as the chaga mushroom. IO commonly grows in Asia, Europe and North America and is widely used as a raw material in various medical conditions [54]. IO has been widely used in traditional medicine to facilitate breathing in Asia and even some parts of Europe, as the mushroom has been known to cause a reduction in nasopharyngeal inflammation [55]. It has been suggested that *Inonotus obliquus* mushrooms possess a powerful enzymatic system and a strong system of defence, due to their parasitic mode of life [56]. Extracts from this fungus have been used for its antitumor, antioxidant, hepatoprotective and anti-inflammatory properties [57]. Moreover, water extracts from IO have traditionally been used as a source of bioactive compounds that exhibit cytostatic and cytotoxic effects, and this has led to the manufacturing of such compounds in the form of a nontoxic aqueous extract called Befungin. Insight into the connection between chaga mushroom and its antiviral effect has been proven to be promising [58]. A study demonstrated the effect of *Inonotus obliquus* polysacharides in cats with feline viruses including feline calcivirus, feline herpesvirus 1, feline influenza virus, feline infectious peritonitis virus and feline panleukopenia virus. Inhibition of RNA viruses and DNA viruses in all the five viral subtypes was observed [58]. Suppression of the infectivity of pandemic influenza virus was also noted in mice and it was observed that this mushroom is comparable to Tamiflu, an antiviral drug that inhibits viral reproduction [59]. Additionally, another study looked at the antiviral effect of *Inonotus obliquus* against (HSV and found that the aqueous extract derived from *I. obliquus* (AEIO) led to an overall reduction in HSV infection in Vero cells. This study also elucidated the anti-HSV mechanism of action, as it found that AEIO had the ability to inhibit viral-induced membrane fusion, thus acting against the early stages of HSV viral infection. Consequently, the results showed that aqueous extracts from *I. obliquus* were able to successfully prevent HSV-1 entry by directly acting on viral glycoproteins, which in turn prevented membrane fusion. Currently, the treatment for HSV infection is a nucleoside analogue antiherpetic. However, increasing resistance to these drugs has resulted in a need to develop alternative treatments. This study has highlighted that the mechanism of action for AEIO is different from that seen in nucleoside analogue antiherpetics and thus provides an alternative treatment to overcome the developing resistance [55].

### Inonotus Obliquus Antiviral, Anti-Inflammatory Effects

Chronic inflammation is the underlying pathogenesis of a series of diseases, including numerous types of carcinomas, atherosclerosis, autoimmune diseases and obesity, among others. Following stimulation caused by lipopolysaccharides (LPS), a series of pro-inflammatory cytokines are released including prostaglandin mediators, cytokines (TNF-α, IL-1β, IL-6) and nitric oxide (NO) [60,61]. It was reported that *IOP* can inhibit the induction of NO and other similar cytokines, a phenomenon that has been associated with COVID-19 [62]. Similarly, in another experiment on inflammatory bowel disease, *Inonotus obliquus* polysaccharides (*IOP)* were shown to alleviate inflammatory responses by inhibiting JAK-STAT signalling pathways that regulate the release of T Helper subsets [63]. Furthermore, the extract of chaga mushroom was also found to possess anticancer properties; however, the exact mechanism of action of these polysaccharides is still unknown [64,65]. Additionally, the effect of *IOP* extracts on Hepatitis C and human immunodeficiency disease have also been elucidated [56,66]. It was concluded that the bioactive molecules of chaga fungus suppressed the expression of JAK-STAT pathway that led to the activation of CD4^+^ T cells, responsible for inflammation [67]. In addition to the mentioned viral diseases, patients infected with COVID-19 also showed similar inflammatory responses possessing significant levels of plasma cytokines and leucocytes. As *IOP*s have shown promising results in treating various viral diseases, the effect of this mushroom in COVID-19 infection could prove to be beneficial.

## 5. Conclusions

Developing viruses such as Ebola virus (EBOV), Lassa virus (LASV), avian influenza virus H5N1 (AIV) and the more recent SARS-COV-2 virus are seen as global health concerns. Despite many advancements in science, no effective vaccine or specific therapy has been approved for humans against these viruses, and so there is an urgent need to develop therapeutic treatments against these threats [68]. Traditional Chinese medicine holds an exclusive position among the variety of traditional medicines because of its thousand-year history. The extracts described in this review have been proven to possess great antiviral activities, with a general consensus of low toxicity. In addition, compared to commercial pharmaceuticals, such medicinal herbs are readily available and much cheaper. With the current pandemic, many scientists have rushed to the development of a potential vaccine and therapeutic agent that is effective against COVID-19; however, herbal agents should not be overlooked. The data presented in this review show the promising effects many herbs and mushrooms have against a variety of viral infections. This review has highlighted the therapeutic potential of *Inonotus obliquus* as a natural antiviral treatment against SARS-COV-2. Earlier studies into this mushroom have laid the groundwork into the antiviral capabilities of *Inonotus obliquus*, however further research into characterising the bioactive ingredients, understanding the underlying mechanisms as well as assessing the efficacy and potential application in vivo should be encouraged in order to develop an effective antiviral treatment against COVID-19. Until the present time, there has not been much research regarding the potential of natural agents against COVID-19, thus opening the research into this field may unlock the potential such extracts may have against SARS-CoV-2. 

## Figures and Tables

**Table 1 nutrients-12-02573-t001:** Antiviral effects of several natural products against coronavirus.

Virus	Natural Product(s) Evaluated	Proposed Mechanism(s)
Coronavirus	Saikosaponins (A, B_2_, C, D) against HCoV-22E9	Saikosaponin B_2_ inhibits viral attachment and penetration stages unclear
	*Lycoris radiata* and its active component lycorine. *Artemisia annua*, *pyrrosia lingua*, and *lindera aggregata* against SARS-CoV1.	
	Phenolic compounds of *Isatis indigotica* against SARS-CoV1.	SARS-CoV1 3CL protease inhibitor
	Amentoflavone isolated from *Torreya nucifera* against SARS-CoV1	SARS-CoV1 3CL protease inhibitor
	Myrcetine and scutellarein against SARS-CoV1	SARS-CoV1 helicase inhibitor
	*Houttuynia cordata water extract* against SARS-CoV1	SARS-CoV1 3CL protease inhibitor; viral polymerase inhibitor

**Table 2 nutrients-12-02573-t002:** The list of introduced medicinal herbs and mushrooms. (+ stands for the level activity, representing mild, moderate, severe and very severe respectively).

Medicinal Herbs and Mushrooms	Antiviral Activity	Anti-Inflammatory Activity	Anticancer Activity
*Prunella vulgaris*	++	+++	++
Garlic (*Allium sativum*)	++	+	++
*Zingiber officinalis*	++	+++	+
*Lentinus edodes* mycelia (shiitake)	+++	+++	-
*Grifola frondosa*	++	+	++
*Ganoderma lucidum* aqueous extract (GLE)	+++	+++	-
*Chlorella vulgaris* ethanolic extract (CVE)	+++	+++	-
*Inonotus obliquus*	++	++++	++++

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
