# Peer review of "The Antiviral, Anti-Inflammatory Effects of Natural Medicinal Herbs and Mushrooms and SARS-CoV-2 Infection"

_nutrients, 2020, doi:10.3390/nu12092573_

Round 1
Reviewer 1 Report
"Najafzadeh et al present a review about “the beneficial effects of natural herbs and mushrooms in COVID- 19 infection”.
Of this paper, first 4,5 pages (up to line 153) are a very nice general review of COVID-19 infection, which could be a separate paper, nevertheless irrelevant of the subject described in the title.
Lines 153-175 start discussing the subject.
Lines 177-239 again going back discussing immunology and pathogenesis of COVID-19 infection, not connected to previous paragraphs
Lines 248-end: discussing again main subject.
We suggest major revisions:
- The title shall be changed because is misleading, as the authors does not provide any evidence for beneficial effects of natural herbs and mushrooms in COVID- 19 infection. They discuss antiviral and antiflammatory effects of natural herbs and mushrooms.
- The structure shall be changed to start with a much shorter introduction including immunology of COVID-19 infection or similar viral infections
- Then the main subject shall be presented"
Author Response
Dear Editor,
The corrections were added to the revised text following the comments:
- The title shall be changed because is misleading, as the authors does not provide any evidence for beneficial effects of natural herbs and mushrooms in COVID- 19 infection. They discuss antiviral and antiflammatory effects of natural herbs and mushrooms.
The title has been changed.
2. The structure shall be changed to start with a much shorter introduction including immunology of COVID-19 infection or similar viral infections.
The structure has been changed according the comment.
3. Then the main subject shall be presented"

Reviewer 2 Report
Line33: the family name .
Line33: Like other CoVs, it is sensitive to ultraviolet rays and heat. REF????
Line46: double‐membrane vesicles (DMVs).6 Subsequently -> 6???
Line74: Viral and host factors that influence the pathogenesis of SARS-CoV-2. Bats are the reservoir of a wide variety of coronaviruses, including severe acute respiratory syndrome coronavirus (SARS-CoV) -like viruses. -> REF??
Line 78: SARS-CoV-2 is an enveloped positive single-stranded RNA (ssRNA) coronavirus -> Again???
Line207: pro-inflammatory cytokines such as IL-7, IL-10, IL-2, G-CSF, MCP-1, MIP-1A and TNFα. -> And IL6
Line 248. It is necessary introduce the JAK-STAT rute
I think the article needs a big table with compound the authors dicuss.
Author Response
Line33: the family name.
Dear Editer,
Thank you for your comments, please find the changes following the comments.
Line33: Like other CoVs, it is sensitive to ultraviolet rays and heat. REF????
The reference has been added.
Line46: double‐membrane vesicles (DMVs).6 Subsequently -> 6???
This was corrected.
Line74: Viral and host factors that influence the pathogenesis of SARS-CoV-2. Bats are the reservoir of a wide variety of coronaviruses, including severe acute respiratory syndrome coronavirus (SARS-CoV) -like viruses. -> REF??
The reference was added.
Line 78: SARS-CoV-2 is an enveloped positive single-stranded RNA (ssRNA) coronavirus -> Again???
The reference was added.
Line207: pro-inflammatory cytokines such as IL-7, IL-10, IL-2, G-CSF, MCP-1, MIP-1A and TNFα. -> And IL6.
This was corrected.
Line 248. It is necessary introduce the JAK-STAT rute
The information was added.
I think the article needs a big table with compound the authors dicuss.
The table was added.
